# Influence of Estrogen Treatment on *ESR1*^+^ and *ESR1*^−^ Cells in ER^+^ Breast Cancer: Insights from Single-Cell Analysis of Patient-Derived Xenograft Models

**DOI:** 10.3390/cancers13246375

**Published:** 2021-12-19

**Authors:** Hitomi Mori, Kohei Saeki, Gregory Chang, Jinhui Wang, Xiwei Wu, Pei-Yin Hsu, Noriko Kanaya, Xiaoqiang Wang, George Somlo, Masafumi Nakamura, Andrea Bild, Shiuan Chen

**Affiliations:** 1Department of Cancer Biology, Beckman Research Institute of the City of Hope, 1500 E Duarte Road, Duarte, CA 91010, USA; hitomi-m@surg1.med.kyushu-u.ac.jp (H.M.); ko-saeki@vet.ous.ac.jp (K.S.); lee.chang.gregory@gmail.com (G.C.); phsu@coh.org (P.-Y.H.); nkanaya77@gmail.com (N.K.); xiaoqiwang@coh.org (X.W.); 2Department of Surgery and Oncology, Graduate School of Medicine, Kyushu University, 3-1-1 Maidashi, Higashi-ku, Fukuoka 812-8582, Japan; mnaka@surg1.med.kyushu-u.ac.jp; 3Integrative Genomics Core, Beckman Research Institute of the City of Hope, 655 Huntington Drive, Monrovia, CA 91016, USA; JinWang@coh.org (J.W.); XWu@coh.org (X.W.); 4Department of Medical Oncology and Therapeutics Research, City of Hope National Medical Center, 1500 E Duarte Road, Duarte, CA 91010, USA; gsomlo@coh.org (G.S.); abild@coh.org (A.B.)

**Keywords:** aromatase inhibitor resistance, ER-positive breast cancer, *ESR1^+^* and *ESR1*^–^ cells, estrogen-induced cell cycle arrest, IL-24, patient-derived tumor xenograft, single-cell RNA sequencing

## Abstract

**Simple Summary:**

The benefit of endocrine therapy is normally observed for cancers with 10% or more of cells positive for ER expression. We compared the gene expression profiles in both *ESR1*^+^ and *ESR1*^–^ cells in ER^+^ tumors following estrogen treatment. Our single-cell RNA sequencing analysis of estrogen-stimulated (SC31) and estrogen-suppressed (GS3) patient-derived xenograft models offered an unprecedented opportunity to address the molecular and functional differences between *ESR1*^+^ and *ESR1*^–^ cells. While estrogen should activate ERα and stimulate *ESR1*^+^ cells, our findings regarding *ESR1*^–^ cells were important, indicating that the proliferation of *ESR1*^–^ cells in ER^+^ cancer is also influenced by estrogen. Another valuable finding from our studies was that estrogen also upregulated a tumor-suppressor gene, *IL-24*, only in GS3. Estrogen increased the percentage of cells expressing *IL-24*, associated with the estrogen-dependent inhibition of GS3 tumor growth.

**Abstract:**

A 100% ER positivity is not required for an endocrine therapy response. Furthermore, while estrogen typically promotes the progression of hormone-dependent breast cancer via the activation of estrogen receptor (ER)-α, estrogen-induced tumor suppression in ER^+^ breast cancer has been clinically observed. With the success in establishing estrogen-stimulated (SC31) and estrogen-suppressed (GS3) patient-derived xenograft (PDX) models, single-cell RNA sequencing analysis was performed to determine the impact of estrogen on *ESR1^+^* and *ESR1*^–^ tumor cells. We found that 17β-estradiol (E2)-induced suppression of GS3 transpired through wild-type and unamplified ERα. E2 upregulated the expression of estrogen-dependent genes in both SC31 and GS3; however, E2 induced cell cycle advance in SC31, while it resulted in cell cycle arrest in GS3. Importantly, these gene expression changes occurred in both *ESR1*^+^ and *ESR1*^–^ cells within the same breast tumors, demonstrating for the first time a differential effect of estrogen on *ESR1*^–^ cells. E2 also upregulated a tumor-suppressor gene, *IL-24*, in GS3. The apoptosis gene set was upregulated and the G2M checkpoint gene set was downregulated in most *IL-24*^+^ cells after E2 treatment. In summary, estrogen affected pathologically defined ER^+^ tumors differently, influencing both *ESR1^+^* and *ESR1*^–^ cells. Our results also suggest *IL-24* to be a potential marker of estrogen-suppressed tumors.

## 1. Introduction

Estrogen plays a crucial role in the progression of hormone-dependent breast cancer via the activation of estrogen receptors (ERs) encoded by *ESR1*. By suppressing estrogen production, aromatase inhibitor (AI) treatments are essential parts of the therapeutic repertoire for ER^+^ postmenopausal breast cancers [1,2,3,4,5,6]. Unexpectedly, several clinical trials have reported a therapeutic benefit of estrogen for AI-resistant advanced breast cancer [7,8,9,10,11,12].

Patient-derived xenografts (PDXs) are superior to cell lines because they typically maintain the biological features of original tumors and have proper multicellular architecture, e.g., ER^+^ tumors contain both ER^+^ and ER^–^ cells. In a previous study, we characterized a trastuzumab-resistant ER^+^/HER2^+^ breast cancer PDX named SC31 [13]. While trastuzumab resistance suggests downregulation of HER2 signaling, estrogen was found to act as a growth driver of SC31 tumors. Whole-genome RNA sequencing (RNA-Seq) transcriptomes and reverse-phase protein array (RPPA) proteome analyses revealed that ERα and mammalian target of rapamycin (mTOR) signaling predominantly regulate tumor growth of SC31. According to dissection of molecular features using omics approaches and prediction analysis of microarray 50 (PAM50) analyses, SC31 (a luminal-A subtype), with high expression of *ESR1* (encoding ERα) and *BCL2*, is an estrogen-stimulation-dependent tumor model [13]. Furthermore, we also succeeded in establishing a unique ER^+^/HER2^–^ PDX model named GS3, derived from an AI-resistant brain metastasis of breast cancer. Our GS3-PDX is a very important model because it is characterized by AI resistance and what turned out to be estrogen-mediated suppression; the specimen for this model was obtained from the patient, rather than being generated in vitro.

The functional characterization of these two ER^+^ PDXs suggests that estrogen can stimulate as well as suppress the progression of ER^+^ cancer. The differential responses to estrogen in SC31 and GS3 provided an opportunity to assess how estrogen modulates ER^+^ cancer. While PDX models are suggested to have “phenotypic drift”, the estrogen responses of SC31 and GS3 are maintained after 14 passages, and have been verified via in vivo estrogen treatment, indicating that these are biologically relevant models. Our investigation confirmed estrogen-stimulated growth of SC31, and revealed a new mechanism of estrogen-mediated suppression in GS3 that is different from those reported using other model systems. Considering the intrinsic heterogeneity in tumors, we performed single-cell RNA sequencing (scRNA-Seq) to evaluate gene expression in both SC31 and GS3 at the individual cell level, and compared the signaling pathways between *ESR1*^+^ and *ESR1*^–^ cells.

## 2. Materials and Methods

### 2.1. PDX

Surgically resected breast cancer tissues were implanted into the 4th mammary fat pad of 6–8-week-old female NOD-SCID/IL2Rγ^−/−^ (NSG) mice to establish the PDX lines. The details for PDX preparation were previously described in [14]. We used two PDX models—SC31 and GS3—for this study. SC31, an estrogen-stimulated model, was established from a trastuzumab-resistant chest wall lesion from a postmenopausal breast cancer patient; the tumor was ER^+^ (80%), progesterone receptor (PR)^−^ (0%), and HER2^+^. Consistent with the clinicopharmacological history of the corresponding patient, an SC31 PDX was derived a from tumor that was already found to be resistant to trastuzumab treatment, with no considerable changes in either tumor volume or weight in our previous study [13]. Although this model is ER^+^/PR^–^/HER2^+^, our previous study demonstrated that the gene expression pattern of SC31 was luminal-A like [13]. GS3—an estrogen-suppressed model—was established from an AI-resistant brain metastasis of a postmenopausal breast cancer patient with the phenotype of ER^+^ (100%), PR^+^ (5%), and HER2^–^. To confirm that GS3 is indeed resistant to AI, NSG mice bearing serially transplanted GS3 were treated with placebo or letrozole (10 mg with 0.3% hydroxypropyl cellulose in 0.9% NaCl solution; daily subcutaneous injection) [15] for 28 days. There was no difference in tumor growth rates or Ki-67 expression (determined by immunohistochemistry (IHC)) between placebo- and letrozole-treated GS3 (Appendix A), while letrozole treatment blocked the murine mammary gland development. Cell viability assay also showed that organoids isolated from the GS3 tumor were resistant to AI (Appendix A).

### 2.2. In Vivo Animal Study

Tumor pieces from established SC31/GS3 lines were implanted into mammary fat pads of 8–10-week-old female ovariectomized/intact NSG mice (GS3 did not grow in ovariectomized NSG mice). Since SC31 tumors grew very slowly without E2, in this study, mice were randomized and implanted with SC31 tumors and pellets (17β-estradiol (E2) 1 mg or placebo) on their abdominal side. For GS3 that could not be established with E2 pellets, mice were randomized after tumor volume reached approximately 200 mm^3^, and were implanted with an E2 (1 mg) or placebo pellet on their backside. For intermittent E2 treatment of GS3, mice with established tumors were implanted with the E2 (1 mg) pellets, which remained in place for 28 days, and then were removed on day 28. After a 28-day interval with no treatment, the mice were again implanted with the E2 (1 mg) pellets for another 28 days. This intermittent E2 treatment was repeated for three rounds. When used, ICI (Fulvestrant, AstraZeneca, Cambridge, UK) was injected subcutaneously (5 mg in 100 μL of sterile saline, once weekly) 4 times. Each treatment group of all animal experiments included at least three mice. All experiments were performed in replicates to confirm that the results that we observed were statistically significant.

### 2.3. Histological Analysis

Hematoxylin and eosin (H&E) staining and IHC of formalin-fixed tumor tissues were performed at the Pathology Core Facility at City of Hope. Antibodies used in IHC included ERα (ab16660, Abcam, Cambridge, UK), PR (PA0312, Leica Biosystems Inc., Wetzlar, Germany), HER2 (A0485, Dako, Glostrup, Denmark), and Ki-67 (M7240, Dako, Glostrup, Denmark). The pathologists evaluated five areas randomly, and ER, PR, and Ki-67 positivity (0–100%) and HER2 IHC score (0–3+) were defined according to the guidelines [16].

### 2.4. Western Blotting

Total protein from PDX tumors was extracted using the Precellys Lysing Kit (Bertin Technologies, Montigny-Le-Bretonneux, France) and lysis buffer (50 mM Tris-HCl, 0.15 M NaCl, 1% Nonidet P40, 0.5% sodium deoxycholate). After centrifuging at 15,000× *g* for 30 min, the supernatants were collected for Western blotting analysis. Antibodies used in Western blotting included ERα (sc-8002, Santa Cruz Biotechnology, Santa Cruz, CA, USA) and GAPDH (#5174, Cell Signaling Technology, Danvers, MA, USA).

### 2.5. RPPA Analysis

Each of the two snap-frozen PDX tumor samples were subjected to RPPA analysis and probed for a total of 232/291 antibodies on SC31 [8]/GS3 (Appendix A) samples. This was conducted by the MD Anderson Cancer Center Functional Proteomics RPPA Facility, as described previously [17].

### 2.6. Real-Time PCR Analysis

Total RNA for real-time PCR was extracted from PDX tumors and organoids using the RNeasy Plus Mini Kit (Qiagen, Hilden, Germany). Reverse transcription reactions were performed with iScript RT reagent (Bio-Rad, Hercules, CA, USA) according to the manufacturer’s instructions. Real-time PCR was performed with SYBER Green FastMix for real-time PCR (Quantabio, Beverly, MA, USA). The mRNA expression was normalized against both β-actin and GAPDH, allowing for comparison of mRNA levels. The primers used in this study are listed in Appendix A.

### 2.7. Bulk RNA-Seq Analysis

Total RNA from PDX tumors was extracted using the RNeasy Plus Mini Kit (Qiagen, Hilden, Germany), and then subjected to RNA-Seq conducted by the Integrative Genomics Core at City of Hope. All RNA samples were extracted from two biological replicates. All sequencing data were submitted to the GEO database. Gene set enrichment analysis (GSEA) was performed using genes ranked by the fold changes between the different conditions to evaluate the significance of activation of 50 hallmark gene sets in MSigDB22.

### 2.8. Single-Cell Preparation

Following a treatment that lasted 6 weeks in SC31 and 7 days in GS3 using E2 (1 mg) or a placebo pellet, tumors were harvested and digested into a single-cell suspension. Although results of longer E2/placebo treatments of GS3 showed larger differences in gene expression according to the bulk RNA-Seq analysis, the viability of single cells isolated from GS3 tumors decreased after the tumors shrank. Since the presence of a large number of dead cells would affect the quality of single-cell analysis, we decided to treat GS3-PDX for 7 days in order to keep single-cell viability over 80%. Single-cell samples were prepared for scRNA-Seq using a 10x Genomics platform. Single-cell preparations from two biological replicates from each treatment were combined and processed for scRNA-Seq (Appendix A). SC31 and GS3 tumors were cut into small, 2 mm thick strips and digested with 1.5 mg/mL DNAse I (#10104159001, Millipore Sigma, St. Louis, MO, USA), 0.4 mg/mL collagenase IV (CLS-4, Lot: 47E17528A, Worthington, Lakewood, NJ, USA), 5% FBS, and 10 mM HEPES in HBSS. The mixture was strained through a 70 μm cell strainer. Then, 1 mL of ACK lysis buffer was used to remove residual red blood cells from the sample. Dead cells were removed using Dead Cell Removal MicroBeads (Miltenyi Biotec, Bergisch Gladbach, Germany).

### 2.9. ScRNA-Seq Analysis

Cell numbers and viability were measured using a TC20 Automated Cell Counter (Bio-Rad, Hercules, CA, USA). We only processed samples that showed at least 80% viability. Cells were then loaded onto the Chromium Controller (10x Genomics, Pleasanton, CA, USA) targeting 2000–5000 cells per lane. The Chromium v2 single-cell 3′ RNA-Seq reagent kit (10x Genomics, Pleasanton, CA, USA) was used to process samples into scRNA-Seq libraries, according to the manufacturer’s protocol. Libraries were sequenced with a HiSeq 2500 instrument (Illumina, San Diego, CA, USA) with a depth of 50,000–100,000 reads per cell. Raw sequencing data were processed using the 10x Genomics Cell Ranger pipeline (version 2.0) to generate FASTQ files, and aligned to the mm10 genome for gene expression count. All sequencing data were submitted to the GEO database. The subsequent data analysis was performed using the Seurat package and R software (version 4.1.1), unless otherwise specified. The custom computer scripts are available in GitHub (https://github.com/HitomiMori/scRNAseq_code_for_R, accessed on 6 December 2021). Cells with mitochondrial read rate >20% and <200 detectable genes were considered to be low quality and were filtered out. After normalization and scaling, cell cycle scoring was performed using the Seurat package, according to the developer’s protocol [18]. Principal component analysis was then performed on 2000 highly variable genes (HVGs). A uniform manifold approximation and projection (UMAP) was generated to summarize and visualize the data in a two-dimensional subspace. Cluster-specific markers were identified in order to generate heat maps and feature plots in the identified cell clusters. GSEA analysis was also performed at the single-cell level, using genes ranked by mean centered log2-normalized read counts and hallmark gene sets in MSigDB.

### 2.10. Organoids and In Vitro Treatment Study

Organoids with the E2-suppressed phenotype were established from PDX-derived surgical specimens of GS3 tumors using 3D culture conditions (Appendix A). After tumor tissue was minced and digested, 10^4^ organoids were embedded in VitroGel 3D-RGD (TWG002, TheWell Bioscience, North Brunswick, NJ, USA) on a 96-well plate and cultured in E2-free M87 medium at 37 °C [19]. Organoid viability was evaluated using the CellTiter-Glo 3D Cell Viability Assay (G9682, Promega, Madison, WI, USA), which measures ATP levels. Each treatment included at least five technical replicates, and we repeated each experiment twice. The ERα-specific antagonist 1,3-bis(4-hydroxyphenyl)-4-methyl-5-[4-(2-piperidinylethoxy)phenol]-1H-pyrazole dihydrochloride (MPP), and the ERβ-specific antagonist, 4-[2-phenyl-5,7-bis(trifluoromethyl)pyrazolo [1,5-β]pyrimidin-3-yl] phenol (PTHPP), were used for co-treatment of E2.

### 2.11. Statistics

To assess statistical significance, values of treated groups were compared to those of control/placebo groups by either two-way ANOVA or Student’s *t*-test, using GraphPad Prism 8 (GraphPad software, San Diego, CA, USA). Error bars represent the SEM. *p*-values of less than 0.05 were considered statistically significant.

## 3. Results

### 3.1. SC31 and GS3 Behaved Oppositely Regarding Tumor Growth with Estrogen

Measurements of tumor volume showed that E2 promoted the growth of SC31 and suppressed the growth of GS3 (Figure 1a). A consistent promotion of SC31 and regression of GS3 were observed in the E2 treatments, in which an E2 pellet (1 mg) was implanted in mice with SC31 (*p* = 0.0006) and GS3 (*p* < 0.0001, Figure 1a). These growth phenotypes were maintained after serial transplantation of SC31/GS3 tumors (up to 13/14 passages so far). H&E staining showed that cell density (the number of cells per microscopic field) increased in SC31 and decreased in GS3 after E2 treatment (Figure 1b). IHC indicated that the proportion of ERα^+^ cells, the ERα staining intensity, and the number of Ki-67^+^ cells increased in SC31 and decreased in GS3 after E2 treatment (Figure 1b). However, IHC showed that PR^+^ cells appeared in both SC31 and GS3 only after E2 treatment (Figure 1b).

### 3.2. E2 Downregulated the Expression of ERα and Cell Cycle Proliferation Genes in GS3

Because the growth responses to E2 in these two ER^+^ breast cancers were unexpectedly different, we performed molecular analyses to determine the mechanism of the different response in GS3, with comparison to the published results [13] on SC31. These analyses included exome sequencing, bulk RNA sequencing, and in vitro validation. Human whole-exome sequencing of GS3 did not detect *ESR1* and *ESR2* variants (Appendix A). The ERα and ERβ genes in both models were wild-type and unamplified [14]. The levels of ERα in two PDXs were comparable, as indicated by RPPA, as published previously [14]. The only other reported estrogen-suppressed PDX model—WHIM16—has an *ESR1* amplification [20,21]. To verify the effect of E2 on the proliferation of GS3 in vitro, organoids were generated from untreated GS3 tumors. E2 suppressed the proliferation of GS3 organoids in an E2-concentration-dependent manner (Figure 2a). These findings demonstrated that estrogen/ERα signaling could modulate ER^+^ tumor growth via atypical mechanisms. To determine which ER subtype was involved in E2-induced regression of GS3, we performed co-treatment of E2 and an ERα-specific antagonist (MPP) or ERβ-specific antagonist (PTHPP) in vitro. E2-mediated inhibition of GS3 organoids could be reversed by the co-treatment with MPP, but not by PTHPP (Figure 2b), indicating the participation of ERα in this suppression process and the reduction in E2-induced suppression by MPP. Western blotting and RPPA analysis showed that the E2 treatment resulted in lower ERα protein levels in vivo (Figure 2c,d and Appendix A). E2 treatment also decreased the *ESR1* expression of GS3 organoids at the mRNA level in vitro (Figure 2e). E2 treatment increased the levels of PR proteins and mRNA (Figure 2d,e, respectively).

To investigate the overall effects of E2 treatment on GS3, bulk RNA-Seq was performed on tumors from mice treated with E2 for 5 and 10 days. GSEA analysis showed that both early and late hallmark estrogen-response gene sets were upregulated after E2 treatment (Appendix A). Although the *ESR1* expression level decreased (possibly in part due to reduction in the number of ERα^+^ cells, as indicated by IHC), expression levels of estrogen-regulated genes—such as *PGR* and *TFF1*—increased after E2 treatment, as expected in estrogen-responsive cells/tumors (Figure 2f). On the other hand, cell cycle progression genes—such as *CDK1*, *TOP2A*, *E2F2*, and *MKI67*—were downregulated in E2-treated GS3 tumors (Figure 2f). The organoids isolated from longer E2-treated tumors expressed estrogen-regulated genes at higher levels (Figure 2g). In addition, the expression of two tumor-suppressor genes—*IL-24* and *GADD45A*—was upregulated in E2-treated PDXs, as well as in organoids that were isolated from E2-treated tumors, depending on treatment time (Figure 2f,g). Collectively, our examination of GS3 tumors after E2 treatment indicates that inhibition of GS3 growth is associated with a reduction in the number of ERα^+^ cells, an increased expression of ERα target genes (indicating the presence of functional active ERα), and the induction of cell cycle arrest and growth suppression.

### 3.3. Impact of E2 on Gene Expression at the Single-Cell Level in SC31 and GS3

To better understand the mechanism of estrogen-regulatory effects, as well as the potential contributions of ERα^+^ cells and ERα^–^ cells within the tumors, we performed scRNA-Seq on individual cells from SC31 and GS3. After removing murine cells from the single-cell suspensions based on species-specific DNA markers (16.2% (SC31) and 10.6% (GS3) of total cells; Appendix A), 19,568 genes were detected from 5581 cells in SC31 (3110 cells from placebo-treated tumors and 2471 cells from E2-treated tumors), while 20,467 genes were detected from 10,631 cells in GS3 (3927 cells from placebo-treated tumors and 6704 cells from E2-treated tumors). A UMAP plot led to the identification of eight major cell clusters in SC31 (Figure 3a) and GS3 (Figure 3b), where each dot represented one cell. The number of cells in each cluster is specified in Appendix A.

### 3.4. Characteristics of Single-Cell Clusters in SC31 and GS3

The relationships between eight clusters of each single-cell analysis were visualized by a heat map using the top 10 differentially expressed genes (DEGs). In SC31, the top 10 DEGs in C2, C4, and C5 included genes belonging to the hallmark G2M checkpoint gene set (Figure 3c, Appendix A). These clusters also included more G2M-phase cells (Appendix A). GSEA analysis showed that the hallmark mTORC1 signaling gene set was upregulated (*p* = 3.30 × 10^−41^) in C6 (Appendix A). In GS3, the top 10 DEGs in C0 and C2 included estrogen-response genes (Figure 3d), and cells in those clusters were mainly G1-phase cells (Appendix A). The hallmark G2M checkpoint gene set was downregulated in these clusters (*p* = 2.02 × 10^−22^ and *p* = 3.76 × 10^−25^, respectively) according to GSEA analysis (Appendix A).

### 3.5. Comparison of E2-Treated Cells vs. Placebo-Treated Cells in SC31 and GS3

In SC31, GSEA analysis of all individual cells showed that the hallmark G2M checkpoint gene set and the hallmark early and late estrogen-response gene sets were significantly upregulated (*p* = 8.99 × 10^−^^72^, 5.43 × 10^−^^52^, 5.43 × 10^−^^52^, respectively) after E2 treatment (Table 1). The hallmark interferon gamma response, interferon alpha response, and TNFA signaling via the NF-κB gene sets were significantly downregulated (*p* = 5.43 × 10^−^^64^, 6.33 × 10^−^^64^, 1.70 × 10^−^^33^, respectively) in E2-treated SC31 cells (Table 1). The percentage of *ESR1*^+^ cells and *PGR*^+^ cells increased by 31% and 37%, respectively, in E2-treated compared to placebo-treated SC31, and the percentage of *MKI67*^+^ cells increased by 15% in E2-treated cells (Figure 4a). In GS3, GSEA analysis of all cells showed that the hallmark early and late estrogen-response gene sets were upregulated (*p* = 1.63 × 10^−^^17^ and 4.55 × 10^−^^16^, respectively) after E2 treatment, while the hallmark G2M checkpoint gene set was downregulated (*p* = 3.95 × 10^−^^13^) (Table 1). The TNFA/NF-κB gene set was also downregulated (*p* = 6.06 × 10^−^^21^) (Table 1). Furthermore, the percentage of *ESR1*^+^ cells decreased by 10%, and the percentage of *PGR*^+^ cells increased by 11%, in E2-treated compared to placebo-treated GS3 (Figure 4b). The percentage of *MKI67*^+^ cells decreased by 4% in E2-treated samples (Figure 4b). The percentage of cells expressing other estrogen-regulated genes—such as *AREG* and *PDZK1*—also increased in individual cells in both E2-treated SC31 and GS3 (Appendix A). The percentage of cells expressing other cell cycle progression genes—such as *CCNE2* and *CDK1*—increased in cells in E2-treated SC31, while it decreased in E2-treated GS3 (Appendix A).

Distribution of cells in the G1, S, or G2/M phases changed after E2 treatment. In SC31, the percentage of cells that progressed to the G2/M phase increased (+6.6%), while the percentage of cells arrested at the G1 phase decreased (−14.5%), in E2-treated samples (Appendix A). In GS3, the percentage of cells that progressed to the G2/M phase decreased (−4.7%), and the percentage of cells arrested at the G1 phase increased (+12.5%), in E2-treated samples (Appendix A). Collectively, the estrogen-regulated genes—such as *PGR*—were expressed after E2 treatment in both SC31 and GS3, while the percentage of cells expressing *ESR1* and *MKI67* or entering the G/2M phase increased in SC31, but decreased in GS3.

For SC31, the proportion of E2-treated cells in C2 and C5 was larger (69% and 65%, respectively) than in other clusters (Appendix A). The percentage of *ESR1*^+^ cells and *PGR*^+^ cells increased in all clusters. Importantly, *MKI67*^+^ cells were mainly present in C2 and C5, and the total number increased after E2 treatment in these two clusters (Appendix A). GSEA analysis also showed that the hallmark G2M checkpoint gene set was upregulated in C2 and C5 when compared to other clusters (Appendix A). For GS3, the proportion of E2-treated cells in C1 was the largest (83%) compared to other clusters (Appendix A). The percentage of *ESR1*^+^ cells was decreased, that of *PGR*^+^ cells was increased, and there were no *MKI67*^+^ cells after E2 treatment in C1 (Appendix A). Similar changes in the distribution of *ESR1*^+^ cells and *PGR*^+^ cells were found in C0. GSEA analysis also showed that the hallmark estrogen-response gene sets were upregulated, and the hallmark G2M checkpoint gene set was simultaneously downregulated in both C0 and C1 when compared to other clusters (Appendix A). Again, *PGR*^+^ cells only appeared in GS3 after E2 treatment (Appendix A). Furthermore, C5 had a higher percentage of *ESR1*^+^ cells as well as *MKI67*^+^ cells, in both placebo and E2-treated samples (Appendix A). Most *MKI67*^+^ cells were at the G2M phase. These observations on C5 indicated that *ESR1*^+^ cells in this cluster functioned differently from those in C0 and C1 of GS3.

### 3.6. Effects of E2 Treatment on ESR1^+^ Cells and ESR1^–^ Cells in SC31 and GS3

The percentages of *ESR1*^+^ and *ESR1*^–^ cells in two tumors treated with E2 or placebo are shown in Appendix A. Since the percentage of *ESR1*^+^ cells changed oppositely in SC31 and GS3 after E2 treatment, we decided to compare the gene expression profiles of *ESR1*^+^ cells and *ESR1*^–^ cells by analyzing them separately. In SC31, the percentage of cells progressing to the G2/M phase increased by 7.2% in *ESR1*^+^ cells and by 5.8% in *ESR1*^–^ cells after E2 treatment. Furthermore, the percentage of cells in the G1 phase decreased by 15.9% and 10.9% in *ESR1*^+^ and *ESR1*^–^ cells, respectively (Figure 4c). In GS3, the percentage of cells arrested at the G1 phase increased by 10.9% in *ESR1*^+^ cells and by 16.0% in *ESR1*^–^ cells after E2 treatment. In addition, the percentage of cells progressing to the G2M phase decreased by 3.4% and 7.8% in *ESR1*^+^ and *ESR1*^–^ cells, respectively. *ESR1*^–^ cells included more G2M-phase cells compared to *ESR1*^+^ cells in placebo-treated GS3 (Figure 4d). However, the decrease in the percentage of G2/M phase cells in GS3 after E2 treatment was greater in *ESR1*^–^ cells compared to *ESR1*^+^ cells (Figure 4d).

To clarify the difference in signaling pathways between *ESR1*^+^ cells and *ESR1*^–^ cells, we performed GSEA analysis for *ESR1*^+^ cells and *ESR1*^–^ cells separately. Firstly, we focused on placebo-treated cells to learn the original characteristics of the two tumors. In SC31, the hallmark G2M checkpoint gene set was upregulated in *ESR1*^+^ cells (*p* = 1.99 × 10^−08^), and the hallmark mTORC1 signaling gene set was upregulated in *ESR1*^–^ cells compared to *ESR1*^+^ cells (*p* = 1.56 × 10^−03^; Table 2). In GS3, the hallmark TNFA/NF-κB signaling gene set was upregulated in *ESR1*^+^ cells (*p* = 4.50 × 10^−34^), and the hallmark G2M checkpoint gene set was upregulated in *ESR1*^–^ cells (*p* = 7.91 × 10^−15^, Table 2). Secondly, we focused on the comparison of placebo-treated vs. E2-treated cells. In SC31, the hallmark G2M checkpoint gene set and early and late estrogen-response gene sets were upregulated, with high statistical significance—not only in *ESR1*^+^ cells (*p* = 1.23 × 10^−72^, 6.52 × 10^−57^, 1.67 × 10^−51^, respectively), but also in *ESR1*^–^ cells (*p* = 1.44 × 10^−48^, 2.49 × 10^−43^, 4.75 × 10^−42^, respectively)—after E2 treatment (Table 3). In GS3, the hallmark early and late estrogen-response gene sets were upregulated not only in *ESR1*^+^ cells (*p* = 1.11 × 10^−19^, *p* = 3.20 × 10^−18^, respectively), but also in *ESR1*^–^ cells (*p* = 1.44 × 10^−15^, *p* = 5.61 × 10^−17^, respectively), after E2 treatment. Moreover, the hallmark G2M checkpoint gene set was significantly downregulated in *ESR1*^–^ cells (*p* = 3.05 × 10^−31^) after E2 treatment (Table 3). Therefore, our single-cell analysis revealed for the first time that E2 treatment upregulated the expression of estrogen-response genes in all epithelial cells of ER^+^ tumors. While estrogen upregulated the expression of cell cycle proliferation genes in both *ESR1*^+^ and *ESR1*^–^ cells of SC31, it downregulated the expression of cell cycle proliferation genes in both types of cells of GS3.

### 3.7. E2-Induced IL-24^+^ Cells through ERα Only in GS3

Interleukin 24 (IL-24) has been shown to be a tumor suppressor [22,23,24,25,26]. In SC31 cells, no *IL-24* expression was observed in either E2-treated or placebo conditions (Figure 5a). In GS3 cells, the percentage of *IL-24*^+^ cells in the E2-treated category was 2.75-fold of that in placebo-treated cells (Figure 5a). *IL-24*^+^ cells were correlated with *ESR1*^+^ cells, especially after E2 treatment (Figure 5b). In *IL-24*^+^ cells, the percentage of cells that progressed to the G2/M phase decreased, and those that arrested at the G1 phase increased when compared to those in the *IL-24*^–^ cells—especially after E2 treatment (Figure 5c). We then checked the levels of *IL-24*^+^ cells in each cluster. The percentage of *IL-24*^+^ cells increased after E2 treatment mainly in C0, C1, C2, and C5 (Figure 5d), with a similar distribution to that seen in *ESR1*^+^ cells in GS3 (Appendix A). The number of G1-phase cells also increased in *IL-24*^+^ cells after E2 treatment in C0, C1, and C2 (Figure 5e). As pointed out previously, C5 contained more *MKI67*^+^ cells (Appendix A). C5 only contained 16% of *IL-24*^+^ cells found in E2-treated GS3, but these cells were in the S and G2M phases (Figure 5e). Further analysis revealed that half of *IL-24*^+^ cells in C5 were positive for *MKI67* (Figure 5f). While these results suggested that few *IL-24*^+^ cells in C5 were proliferative, the majority of *IL-24*^+^ cells in E2-treated GS3 were associated with G1 arrest (Figure 5f).

When comparing *IL-24* expression levels in total mRNA extracted from GS3 tumors, *IL-24* expression levels were upregulated after E2 treatment in a time-dependent manner, and *IL-24* expression was inhibited by ICI (Figure 5g). To clarify the difference in signaling pathways between *IL-24*^+^ and *IL-24*^–^ cells, we performed GSEA analysis for *IL-24*^+^ and *IL-24*^–^ cells separately. The hallmark TNFA/NF-κB signaling, hallmark early and late estrogen response, and hallmark apoptosis gene sets were upregulated in *IL-24*^+^ cells after E2 treatment (*p* = 1.33 × 10^−17^, 1.73 × 10^−9^, 1.73 × 10^−9^, 2.45 × 10^−6^, respectively). On the other hand, the hallmark G2M checkpoint gene set was downregulated in *IL-24*^+^ cells after E2 treatment (*p* = 8.25 × 10^−9^, Table 4).

### 3.8. Intermittent E2 Treatment in GS3

To investigate possible changes in response to E2 in GS3 tumors, we exposed PDX mice to cycles of intermittent E2 treatment (on and off every 28 days). A repeating pattern of tumor growth inhibition (E2-exposed) and growth (E2-absent) was initially observed. However, after three rounds of intermittent E2 treatment on GS3, an E2-independent growth developed (Figure 6a). The decrease in the numbers of ERα^+^ cells and Ki-67^+^ cells evaluated after E2 treatment was reversed after three rounds of intermittent E2 treatment (Figure 6b), as were ERα protein levels as measured by Western blotting (Figure 2c). The increase in the number of PR^+^ cells by IHC after E2 treatment was not reversed after intermittent E2 treatment (Figure 6b). To identify the key pathway driving this emerging E2 independence, we performed bulk RNA-Seq on GS3 tumors treated with placebo, E2 (for 28 days), or intermittent E2. The hierarchical clustering using 1312 DEGs showed that approximately 60% of genes in the intermittent-E2-treated sample had the same trend as E2-treated samples (mainly estrogen-regulated genes), in which 40% of the genes behaved similarly to the placebo-treated sample (mainly cell cycle progression genes) (Figure 6c). After E2 treatment for 28 days, the hallmark late estrogen-response gene set was upregulated, and the hallmark G2M checkpoint gene set was downregulated, consistent with the results of scRNA-Seq analysis (Appendix A). Interestingly, the hallmark reactive oxygen species (ROS) pathway gene set was upregulated after the long-term E2 treatment, despite the effective pathway size being relatively small (Appendix A). Upregulation of *IL-24* expression level and the hallmark ROS pathway after 28-day exposure to E2 was reversed in intermittent-E2-exposed samples (Figure 5g, Appendix A), supporting the role of *IL-24* in GS3. Downregulation of the hallmark G2M checkpoint gene set after 28-day exposure to E2 was also reversed in intermittent-E2-exposed samples (Appendix A).

## 4. Discussion

Through scRNA-Seq of two ER^+^ PDX tumors, we could analyze the gene expression profiles of individual *ESR1*^+^ and *ESR1*^–^ cells from ER^+^ breast cancers with or without estrogen treatment. Clinical data on the subgroup of cancers with ER-low-positive (1−10%) were less definitive [27], but 100% ER positivity is not required for an endocrine therapy response. However, there has been little information about ER^–^ cells in endocrine-responsive tumors.

Our scRNA-Seq analysis of SC31 and GS3 offered an unprecedented opportunity to address the molecular and functional differences between *ESR1*^+^ and *ESR1*^–^ cells in ER^+^ tumors that were responsive to estrogen treatment. After analyzing DEGs in *ESR1*^+^ and *ESR1*^–^ cells separately, the G2M checkpoint gene set and early and late estrogen-response gene sets in SC31 were found to be upregulated not only in *ESR1*^+^ cells, but also in *ESR1*^–^ cells. In GS3, the G2M checkpoint gene set was downregulated mainly in *ESR1*^–^ cells, and the E2 response gene sets were upregulated in both *ESR1*^+^ and *ESR1*^–^ cells. This means that *ESR1*^–^ cells were also affected by E2, and behaved similarly to *ESR1*^+^ cells. We compared the percentages of *ESR1*^+^ and *ESR1*^–^ cells isolated from placebo- and E2-treated PDXs (Appendix A). Our results further support the hypothesis that E2 affects both *ESR1*^+^ and *ESR1*^–^ cells. If only *ESR1*^+^ cells were affected, we would expect to find mainly *ESR1*^+^ cells in SC31 and *ESR1*^–^ cells in GS3 after E2 treatment. Furthermore, in our analysis, the activation of the mTOR signaling pathway in SC31 was observed in *ESR1*^–^ cells. Our findings [13] on the combined effect of ICI and mTOR inhibitors on the SC31 model further support the case for estrogen-related effects on *ESR1*^–^ cells in this tumor model. While the detailed mechanisms of the potential interactions between *ESR1*^+^ cells and *ESR1*^–^ cells are yet to be better defined, this finding offers an important explanation as to why most of the estrogen-responsive breast cancers are not homogeneously 100% ER^+^. Furthermore, our results explain why it is clinically unusual that endocrine therapy of ER^+^ cancer results in a total loss of ER positivity in the residual tumors.

SC31 is a trastuzumab-resistant, ER^+^/HER2^+^ PDX model with gene expression characteristic of luminal-A breast cancer. As shown previously [28,29], E2 signaling increased the expression of estrogen-regulated genes and genes involved in cell proliferation (Figure 4a and Appendix A). PI3K/AKT/mTOR signaling is a critical oncogenic pathway, and ERα signaling plays a dominant role in SC31 [13]. ER-mediated pathways are activated in ER^+^/HER2^+^ cancers [30], and ER regulation becomes dominant after HER2 signaling is no longer the driver. SC31 responded to E2 in the same manner as typical luminal-A ER^+^ breast cancer.

AIs are part of adjuvant and advanced disease therapy for luminal-A-type postmenopausal breast cancer. Development of AI resistance is not uncommon [5]. Interestingly, estrogen-induced tumor regression has been reported in some AI-resistant ER^+^ breast cancer cases [7,8,9,10,11,12]. Considering that AI-resistant cancers develop an ability to grow without estrogen, AI-resistant cell lines have been established from hormone-dependent breast cancer cell lines inclusive of MCF-7 and T-47D cells, via culture under long-term estrogen deprivation (LTED) conditions [31,32,33,34,35]. Some of these LTED cell lines showed higher ER expression [35] and estrogen hypersensitivity [31]. In some LTED cell lines, growth factor receptors and downstream signaling pathways—including the MAPK, PI3K/AKT/mTOR, and JNK pathways—could function as alternative growth signaling pathways, independent of ER [36], or via ER–growth factor coregulation mechanisms [37]. Cell lines and a PDX model have been used to investigate the mechanism of varying responses to E2 in breast cancer [20,38,39,40,41,42]. In our investigations of the SC31 and GS3 PDX models by scRNA-Seq, we obtained new and valuable information on how E2 suppresses the growth of patient-derived tumors in the murine PDX model. While, as expected, E2 exposure increased the expression of ERα-regulated genes in the GS3 model, our single-cell analyses revealed that E2 reduced the proportion of cells expressing genes involved in cell proliferation (Figure 4b and Appendix A). While others have suggested that AI resistance is caused by mutation [43,44] or amplification [20,21] of *ESR1*, our results for GS3 indicated that there are additional mechanisms. AI-resistant tumors often retain ER expression and ER signaling [45]. ER is constitutively active in GS3, as reported for AI-resistant tumors [45]. To verify the important role of ER in GS3, in vivo treatment of placebo/E2/ICI/E2+ICI for GS3 was performed. ICI is an ER degrader that reduces ER levels, and is different from MPP, which is an ER antagonist. E2, ICI, and E2+ICI treatments significantly suppressed GS3 tumor growth compared to placebo treatment (Appendix A). These results confirm that ER is essential for tumor growth, and that GS3 without E2 has constitutively active ER (possibly linked to the TNF/NF-κB pathways) [46,47]. While others have also suggested that the mechanism of E2-induced tumor suppression includes unfolded protein response (UPR) [20], our scRNA-Seq analysis did not find the hallmark UPR gene set to be upregulated in GS3 after E2 treatment.

Our study revealed that the hallmark interferon gamma response, interferon alpha response, and TNFA/NF-κB gene sets were significantly downregulated in both *ESR1*^+^ and *ESR1*^–^ cells after E2 treatment (Table 3). These results indicate potential negative associations between estrogen signaling and immune/inflammation responses. De Angelis et al. [48] reported that the activation of these pathways was associated with intrinsic and acquired resistance to CDK4/6 inhibitors, implying mechanisms of dysregulation of ER signaling associated with resistance to these cell cycle inhibitors.

Importantly, the results of bulk RNA-Seq of GS3 tumors revealed that the *IL-24* expression level increased markedly after E2 treatment, in a time-dependent manner (Figure 2f); we confirmed that *IL-24* was upregulated by ERα in GS3 (Figure 5g). *IL-24* was not included in the hallmark early and late estrogen-response gene sets, but it has been documented that *IL-24* was upregulated by E2 preferentially through ERα [49]. IL-24 is a tumor-suppressor cytokine that selectively induces cell cycle arrest and apoptosis in a wide variety of human cancer cells, including breast cancer [22,23,24,25,26]. *IL-24*^+^ cells were arrested in the G1 phase of the cell cycle (Figure 5c), downregulating the hallmark G2M checkpoint gene set, and upregulating the hallmark apoptosis gene set (Table 4). Furthermore, upregulation of *IL-24* expression levels and the hallmark ROS pathway after 28-day exposure to E2 was reversed in intermittent E2-exposed GS3 (Figure 5g and Appendix A), which resulted in E2-independent growth of the tumors. Therefore, our results suggest that estrogen-mediated suppression of GS3 takes place through its induction of *IL-24* expression vs. ER. Publicly available data showing that high expression of IL-24 at the protein level is related to longer overall survival in breast cancer patients also further supported our results (*p* = 0.00051; Appendix A). Our analysis revealed that TNFA/NF-κB gene set has a strong association with *IL-24*^+^ cells. NF-κB regulates the transcription of a large array of cytokines and chemokines, and also induces IL-24 production [50]. The NF-κB pathway has also been linked to endocrine therapy resistance, and has been suggested to be associated with ER phosphorylation (at S305), i.e., activation of ER without estrogen. [46,47].

Estrogen-induced cell cycle arrest can be an unexpected sequence in AI resistance, suggesting that estrogen therapy may be viable for patients who fail AI therapy. In ER^+^ breast cancer, estrogen is typically assumed to promote tumor growth. Therefore, clinicians hesitate to use E2 as a general treatment option for recurrent ER^+^ tumors, even when the tumors are completely resistant to AI. Our mechanistic studies of GS3 offer leads for identifying tumors that can respond to E2 therapies. The clinical benefit rate of estrogen therapies for AI-resistant cases was 26–56% in 7 trials (Appendix A). Our findings point out that measurements of ER and PR expression alone are not sufficient to propose E2 therapy. Our study shows that IL-24 is a biomarker candidate for selecting patients and predicting the effects of E2 treatment in AI-resistant ER^+^ postmenopausal breast cancer.

We recognize several limitations to this study, including the technical limitations involved in isolating 100% living cells after tumor digestion for single-cell analysis. Furthermore, we observed differences between the ERα positivity rate of placebo-treated SC31/GS3 at the protein level by IHC and the *ESR1* positivity rate at the mRNA level. The isolation of single cells from solid tissue remains empirical. It is likely that different types of cells are fractionated with different efficiency. Although we cannot confirm this directly in this study, there might be some *ESR1*^–^ cells that express functional ERα because of the time lags of the protein turnover. This could be one mechanism that explains why *ESR1*^–^ cells responded to E2 treatment. An additional limitation is that our study included a single estrogen-suppressed PDX model. There are only two estrogen-suppressed PDXs to date—GS3 and WHIM16 [20,21]. We consider GS3 to be a valuable tumor model derived from a patient, which reveals a new mechanism of estrogen-induced cell cycle arrest, indicating that there may be more than one means of estrogen-mediated suppression of breast cancer. With this in mind, additional estrogen-suppressed tumor specimens will be needed in order to verify these mechanisms. Furthermore, we recognize that the limitation of the scRNA-Seq analysis is its computational nature, and that the existing data are mainly correlative. While it is significant to emphasize that we have been able to directly compare the E2-induced changes in the gene expression profiles of two PDXs with definitive growth responses in vivo, nevertheless, more functional studies will be needed in order to determine whether changes in the expression of *IL-24* (and other genes) are causally linked to E2-mediated tumor changes.

## 5. Conclusions

Estrogen/ERα signaling increases the expression of estrogen-regulated genes (e.g., *PGR* and *AREG*). E2 promotes ER^+^ breast cancer growth, as seen in the case of SC31, while E2-induced suppression is an unexpected outcome of AI resistance (such as in GS3). In the latter case, elimination of estrogen by AI results in maintaining tumor growth. Analysis of GS3-PDX has revealed that estrogen can induce cell cycle arrest. As a conceptual advancement, our results reveal potential interactions between *ESR1^+^* and *ESR1*^–^ cells in both estrogen-stimulated and -suppressed ER^+^ tumors, as well as a potential tumor-suppressor role of *IL-24*. Our findings point to the need to identify biomarkers for patients with estrogen-suppressed tumors who may benefit from E2 treatment after AI resistance. Expression of *IL-24* in AI-resistant tumors may be one such indicator of favorable response to E2 therapy.

## Figures and Tables

**Figure 1 cancers-13-06375-f001:**
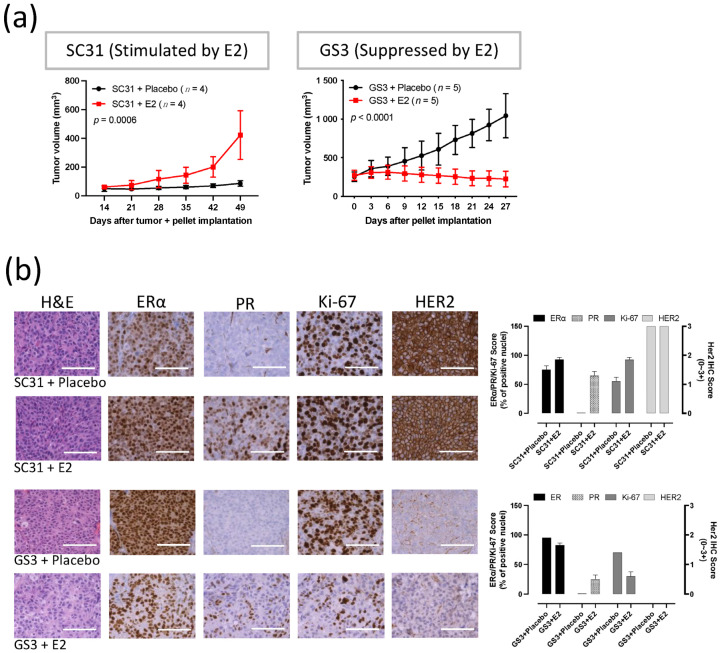
E2 promoted the growth of SC31 and inhibited the growth of GS3: (**a**) Tumor growth curve of SC31/GS3-PDX with E2 (1 mg) or placebo treatment. Tumors and pellets were implanted at the same time for SC31 using OVX mice, and pellets were implanted in intact mice after tumor volume reached approximately 150–200 mm^3^ for GS3; *p*-values were determined by two-way ANOVA analysis. (**b**) Hematoxylin and eosin staining and immunohistochemistry of SC31/GS3 tumors treated with placebo or E2 (1 mg). Scale bar represents 100 μm. Error bars represent SEM.

**Figure 2 cancers-13-06375-f002:**
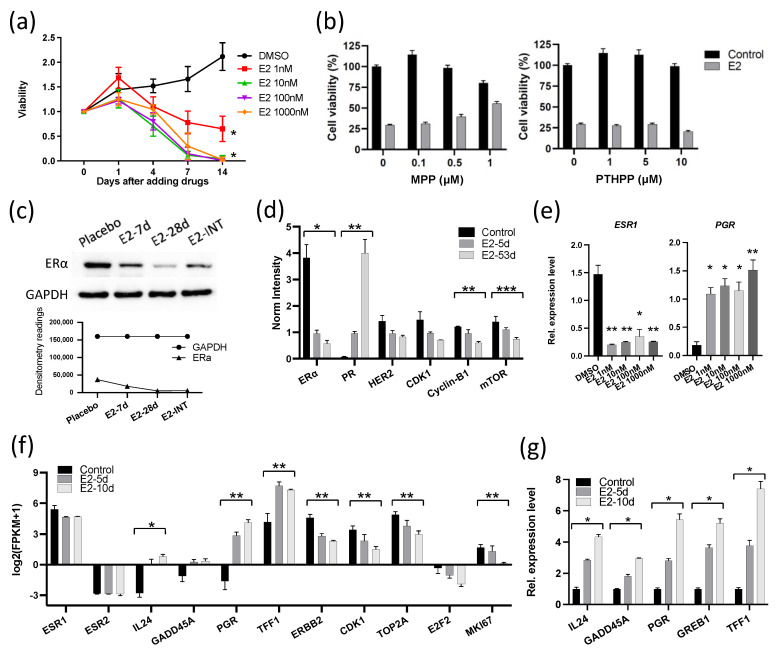
E2 suppressed the expression of ERα and cell cycle proliferation genes in GS3: (**a**) Cell viability of 10^4^ organoids (isolated from GS3 tumor) treated with varying concentrations of E2 in vitro (* *p* < 0.0001). (**b**) Cell viability of organoids (isolated from GS3 tumor) co-treated with E2 and an ERα antagonist (MPP) or ERβ antagonist (PTHPP) in vitro. (**c**) ERα expression of GS3 tumors treated with placebo, 7-day E2, 28-day E2, and intermittent E2 by Western blotting; whole blots can be found in Appendix A. (**d**) Protein expression levels of GS3 tumors treated with 5- and 53-day E2, and control samples, as determined by reverse-phase protein array analysis (* *p* = 0.013, ** *p* < 0.01, *** *p* = 0.04). (**e**) *ESR1* and *PGR* expression levels of GS3 organoids treated with varying concentrations of E2 in vitro (* *p* < 0.05, ** *p* < 0.01). (**f**) Log2 (fpkm + 1) of gene expression in GS3 tumors treated with 5- and 10-day E2, and control samples, as determined by bulk RNA-Seq (* *p* = 0.007, ** *p* < 0.05). (**g**) Gene expression levels of GS3 organoids isolated from GS3 tumors with 5- and 10-day E2, and control samples (* *p* < 0.0001). Error bars represent the SEM, and *p*-values were determined by two-way ANOVA analysis (**a**) and multiple unpaired *t*-tests (**d**–**g**). E2-INT: intermittent E2 treatment.

**Figure 3 cancers-13-06375-f003:**
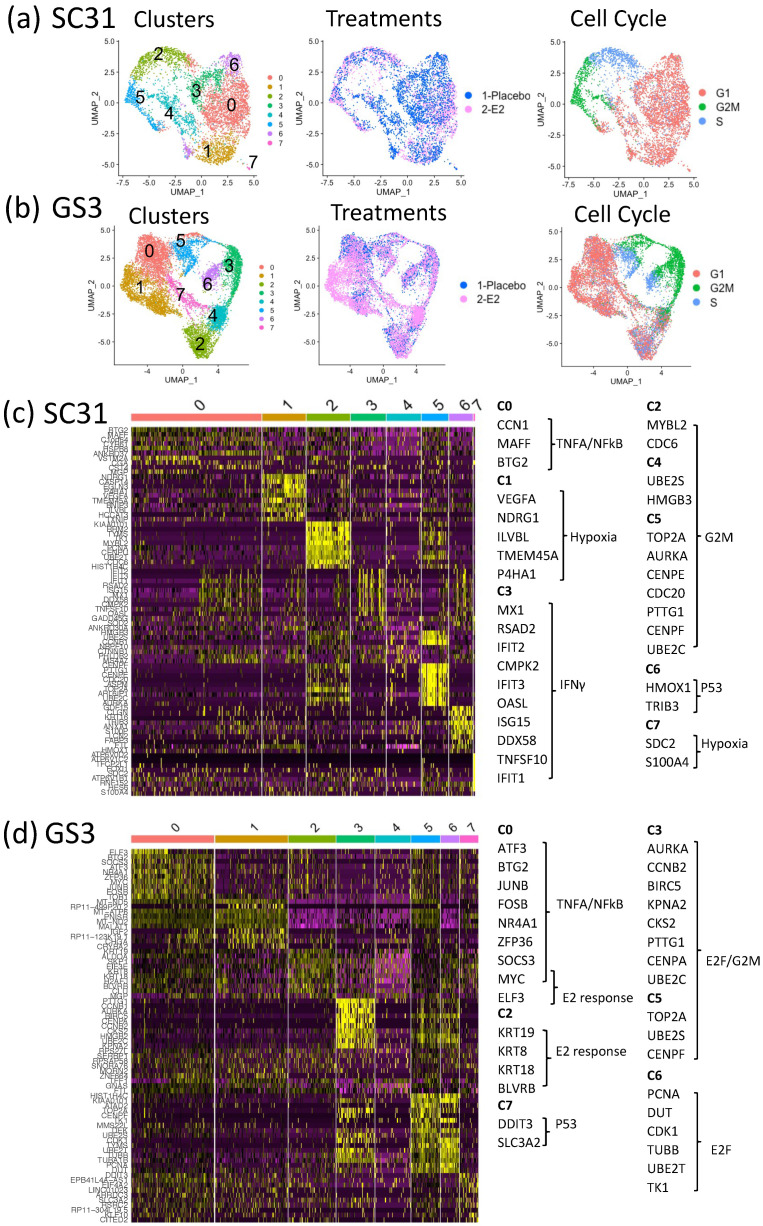
SC31 and GS3 tumors were analyzed by scRNA-Seq: (**a**) UMAP of cells from both placebo- and E2-treated SC31-PDX, color-coded according to clusters, treatments, and cell cycle phases. (**b**) UMAP of cells from both placebo- and E2-treated GS3-PDX, color-coded according to clusters, treatments, and cell cycle phases. (**c**) Heat map using the top 10 differentially expressed genes in SC31. (**d**) Heat map using the top 10 differentially expressed genes in GS3.

**Figure 4 cancers-13-06375-f004:**
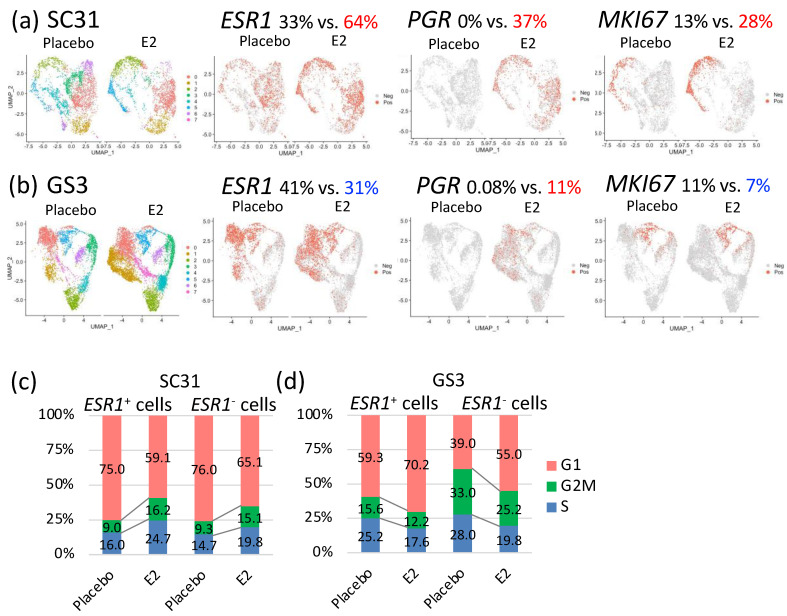
Gene expression comparison of cells from tumors treated with E2 vs. placebo: selected feature plots of cells expressing *ESR1*, *PGR*, and *MKI67* in UMAP, separated by treatments in (**a**) SC31 and (**b**) GS3; distribution of cells in the G1, S, or G2/M phases in *ESR1*^+^ and *ESR1*^–^ cells, separated by treatments in (**c**) SC31 and (**d**) GS3.

**Figure 5 cancers-13-06375-f005:**
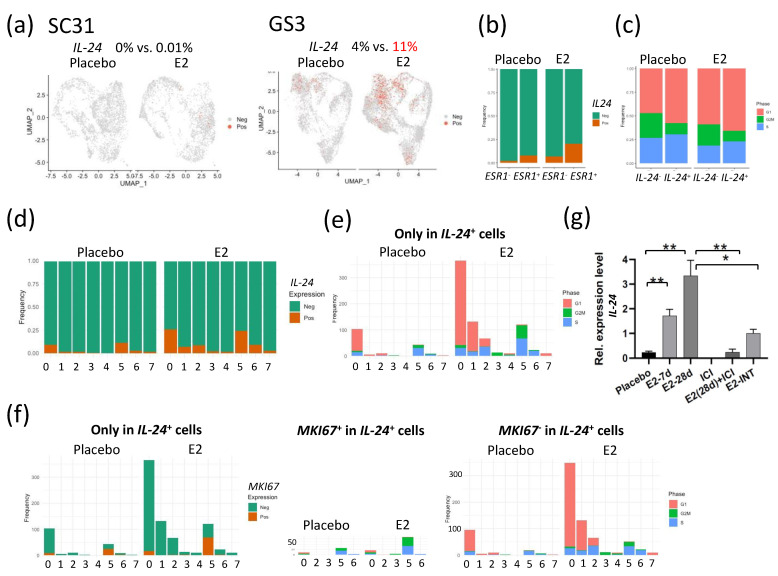
E2 treatment of tumors altered the expression of *IL-24* in individual cells: (**a**) Feature plot of cells expressing *IL-24* in UMAP in SC31 and GS3. (**b**) Distribution of cells expressing *IL-24* in *ESR1*^+^ or *ESR1*^–^ cells separated by treatments in GS3. (**c**) Distribution of cells in the G1, S, or G2/M phases in *IL-24*^+^ or *IL-24*^–^ cells, separated by treatments. (**d**) Cell distribution of single-cell clusters by *IL-24* expression (separated by treatments) in GS3. (**e**) Distribution of cells in the G1, S, or G2/M phases, only in *IL-24*^+^ cells, separated by treatments, in GS3. (**f**) Cell distribution of single-cell clusters by *MKI67* expression, and distribution of cells in the G1, S, or G2/M phases, separated by *MKI67* expression, only in *IL-24*^+^ cells, separated by treatments, in GS3. (**g**) *IL-24* expression levels in GS3 tumors with different treatments, including E2 and ICI (* *p* < 0.05, ** *p* < 0.01). Error bars represent the SEM, and *p*-values were determined via multiple unpaired *t*-tests. E2-7d: E2 treatment for 7 days; E2-28d: E2 treatment for 28 days; ICI: ICI treatment for 28 days (once weekly, four times); E2-INT: intermittent E2 treatment.

**Figure 6 cancers-13-06375-f006:**
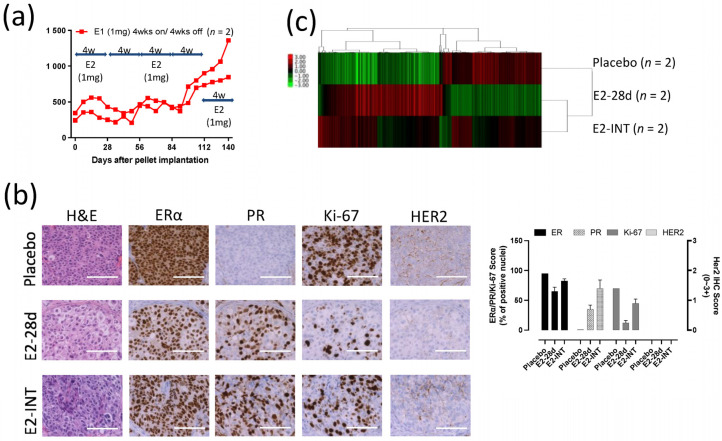
Intermittent E2 treatment in GS3: (**a**) Tumor growth curves of intermittent E2 (1 mg) treatment that was cycled every 28 days (E2 pellets on/off every 4 weeks, three times), after tumor volume reached approximately 200 mm^3^. (**b**) Hematoxylin and eosin staining and immunohistochemistry of GS3 tumors treated with placebo, E2 (4 weeks), or intermittent E2; scale bar represents 100 μm. (**c**) Hierarchical clustering using 1312 differentially expressed genes from bulk RNA-Seq.

**Table 1 cancers-13-06375-t001:** Hallmark gene sets analysis of SC31 and GS3 (E2-treated vs. placebo).

ER+ BC Model	Upregulated in E2	*p*-Value ^1^	Downregulated in E2	*p*-Value ^1^
SC31	E2F_TARGETS	5.94 × 10^−^^84^	INTERFERON_GAMMA_RESPONSE	5.43 × 10^−^^64^
	MYC_TARGETS_V1	2.88 × 10^−^^76^	INTERFERON_ALPHA_RESPONSE	6.33 × 10^−^^64^
	G2M_CHECKPOINT	8.99 × 10^−^^72^	TNFA_SIGNALING_VIA_NFKB	1.70 × 10^−^^33^
	ESTROGEN_RESPONSE_EARLY	5.43 × 10^−^^52^	APOPTOSIS	6.64 × 10^−^^25^
	ESTROGEN_RESPONSE_LATE	5.43 × 10^−^^52^	P53_PATHWAY	2.15 × 10^−^^18^
GS3	ESTROGEN_RESPONSE_LATE	1.63 × 10^−^^17^	TNFA_SIGNALING_VIA_NFKB	6.06 × 10^−^^21^
	ESTROGEN_RESPONSE_EARLY	4.55 × 10^−^^16^	G2M_CHECKPOINT	3.95 × 10^−^^13^
			P53_PATHWAY	3.95 × 10^−^^13^
			APOPTOSIS	5.90 × 10^−^^12^
			E2F_TARGETS	6.23 × 10^−^^12^

^1^ This table includes the top five hallmark gene sets, or those whose *p*-value was smaller than 1.0 × 10^−10^.

**Table 2 cancers-13-06375-t002:** Analysis of hallmark gene sets of *ESR1^+^* cells vs. *ESR1*^–^ cells in placebo or E2.

	Upregulated in *ESR1*^+^ Cells	*p*-Value ^1^	Upregulated in *ESR1*^−^ Cells	*p*-Value ^1^
**SC31**				
Placebo	G2M_CHECKPOINT	1.99 × 10^−08^	HYPOXIA	2.20 × 10^−09^
	E2F_TARGETS	8.22 × 10^−07^	GLYCOLYSIS	1.56 × 10^−03^
	TNFA_SIGNALING_VIA_NFKB	7.00 × 10^−04^	MTORC1_SIGNALING	1.56 × 10^−03^
E2	ESTROGEN_RESPONSE_EARLY	1.03 × 10^−30^	HYPOXIA	3.64 × 10^−13^
	ESTROGEN_RESPONSE_LATE	3.42 × 10^−21^	GLYCOLYSIS	1.08 × 10^−09^
	G2M_CHECKPOINT	9.81 × 10^−07^	MTORC1_SIGNALING	4.61 × 10^−08^
**GS3**				
Placebo	TNFA_SIGNALING_VIA_NFKB	4.50 × 10^−34^	MYC_TARGETS_V1	1.44 × 10^−34^
	APOPTOSIS	1.45 × 10^−12^	E2F_TARGETS	1.20 × 10^−29^
	ESTROGEN_RESPONSE_EARLY	1.54 × 10^−11^	G2M_CHECKPOINT	7.91 × 10^−15^
E2	TNFA_SIGNALING_VIA_NFKB	7.22 × 10^−11^	MYC_TARGETS_V1	4.89 × 10^−12^
	UV_RESPONSE_UP	1.80 × 10^−04^	E2F_TARGETS	4.17 × 10^−08^
			G2M_CHECKPOINT	2.68 × 10^−06^

^1^ This table includes the top three hallmark gene sets.

**Table 3 cancers-13-06375-t003:** Analysis of hallmark gene sets of *ESR1^+^* cells and *ESR1*^–^ cells (E2-treated vs. placebo).

	Upregulated in E2	*p*-Value ^1^	Downregulated in E2	*p*-Value ^1^
**SC31**				
*ESR1*^+^ cells	E2F_TARGETS	2.91 × 10^−83^	INTERFERON_GAMMA_RESPONSE	6.11 × 10^−64^
	G2M_CHECKPOINT	1.23 × 10^−72^	INTERFERON_ALPHA_RESPONSE	6.81 × 10^−63^
	MYC_TARGETS_V1	1.10 × 10^−69^	TNFA_SIGNALING_VIA_NFKB	1.06 × 10^−34^
	ESTROGEN_RESPONSE_EARLY	6.52 × 10^−57^	APOPTOSIS	2.82 × 10^−23^
	ESTROGEN_RESPONSE_LATE	1.67 × 10^−51^	P53_PATHWAY	3.16 × 10^−21^
*ESR1*^–^ cells	E2F_TARGETS	1.94 × 10^−62^	INTERFERON_ALPHA_RESPONSE	4.80 × 10^−66^
	MYC_TARGETS_V1	1.94 × 10^−62^	INTERFERON_GAMMA_RESPONSE	7.63 × 10^−66^
	G2M_CHECKPOINT	1.44 × 10^−48^	TNFA_SIGNALING_VIA_NFKB	7.61 × 10^−34^
	ESTROGEN_RESPONSE_EARLY	2.49 × 10^−43^	APOPTOSIS	3.63 × 10^−25^
	ESTROGEN_RESPONSE_LATE	4.75 × 10^−42^	HYPOXIA	3.83 × 10^−16^
**GS3**				
*ESR1*^+^ cells	ESTROGEN_RESPONSE_EARLY	1.11 × 10^−19^	TNFA_SIGNALING_VIA_NFKB	2.20 × 10^−22^
	ESTROGEN_RESPONSE_LATE	3.20 × 10^−18^	P53_PATHWAY	3.69 × 10^−13^
			APOPTOSIS	5.57 × 10^−12^
*ESR1*^–^ cells	ESTROGEN_RESPONSE_LATE	5.61 × 10^−17^	G2M_CHECKPOINT	3.05 × 10^−31^
	ESTROGEN_RESPONSE_EARLY	1.44 × 10^−15^	E2F_TARGETS	8.25 × 10^−23^
			TNFA_SIGNALING_VIA_NFKB	5.48 × 10^−14^

^1^ This table includes the top five hallmark gene sets, or those whose *p*-value was smaller than 1.0 × 10^−10^.

**Table 4 cancers-13-06375-t004:** Analysis of GSEA hallmark gene sets of genes expressed in *IL-24^+^* cells vs. *IL-24*^–^ cells.

Upregulated in *IL24*^+^ Cells	*p*-Value ^1^	Downregulated in *IL24*^+^ Cells	*p*-Value ^1^
TNFA_SIGNALING_VIA_NFKB	1.33 × 10^−17^	G2M_CHECKPOINT	8.25 × 10^−9^
ESTROGEN_RESPONSE_ EARLY	1.73 × 10^−9^	E2F_TARGETS	8.25 × 10^−9^
ESTROGEN_RESPONSE_ LATE	1.73 × 10^−9^	MYC_TARGETS_V1	5.24 × 10^−5^
APOPTOSIS	2.45 × 10^−6^	P53_PATHWAY	2.46 × 10^−3^

^1^ This table includes all hallmark gene sets from the results of GSEA.

## Data Availability

The raw bulk RNA-Seq and scRNA-Seq data generated in this study have been deposited in GEO. SC31; scRNA-Seq, https://www.ncbi.nlm.nih.gov/geo/query/acc.cgi?acc=GSE176532, accessed on 10 June 2020; GS3; bulk RNA-Seq, https://www.ncbi.nlm.nih.gov/geo/query/acc.cgi?acc=GSE156922, accessed on 9 September 2020; GS3; scRNA-Seq, https://www.ncbi.nlm.nih.gov/geo/query/acc.cgi?acc=GSE156752, accessed on 24 August 2020.

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
