# Peer review of "Influence of Estrogen Treatment on ESR1+ and ESR1 Cells in ER+ Breast Cancer: Insights from Single-Cell Analysis of Patient-Derived Xenograft Models"

_cancers, 2021, doi:10.3390/cancers13246375_

Round 1

Reviewer 1 Report

This study shows the response and alteration of breast cancer PDX models with E2 treatment.
The authors establish two breast cancer PDX models, and these models have the difference response against E2 treatment.
These differences were discovered by expressional analysis with bulk scale and single-cell scale.
Their concepts are interesting; however, there are several concerns for the publication.

Comments
1. The authors should show the analysis code. And the version of analysis soft should be shown in the materials and methods section.

2. In Figure 1B, the scale bars are difficult to see. It should be more clear.

3. In Figure 2D, the authors shows the protein array of PDX tumor. The raw data should be available to access. Additionally, this results concluding the expression of stromal cells derived from human(patient) and mouse(NSG). Do these alterations specifically occur in tumor cell? 

4. In Figure 2F, probably, log2(FPKM+1) is correct. In bulk RNA-seq analysis, 

5. In single-cell analysis, the authors showed the alteration by E2 treatment. The authors hypothesized the IL24 relation in E2 treatment in Figure 7. However, this is an overstatement. First, Estrogen signaling and E2 treatment deeply connected to cell cycle and cell proliferation. So, generally, these cell alteration of population has been induced by cell proliferation. Therefore, the results in Figure S5 are important to understand the dynamics by E2 treatment. The authors should analyze the cell cycle and DEGs between placebo and E2 treatment in each cluster.
Which cluster is the most affected by E2? And is the IL24 expression altered in the most affected cluster? The authors should perform the more detailed analysis in scRNA-seq.

6. In single-cell analysis, are there the E2 treatment-specific clusters? If there is no specific clusters, it can be considered that the cell number alteration causes the  cell population changes in the PDX models.

7. PDX models has the stromal cells derived from patient and host mice. Therefore, scRNA-seq enables to detect the several types of stroma. However, the authors shows the only cancer cell cluster. The authors should show and describe the detail of cluster identification in single-cell analysis. What the markers of cancer cell are used in the analysis?

8. In Figure 5, the authors should analyze the IL24 expression in each cluster. If the authors analyze it as a whole, it is the same as bulk analysis.

Reviewer 2 Report

This paper by Mori et al. described an interesting yet poorly understood clinical phenomenon of tumor inhibition by estrogen of certain ESR1+ breast cancer. The authors leveraged a recently developed PDX model (GS3) from a breast cancer patient with such clinical manifestation, and compared it to a previously developed ESR1+ breast cancer PDX model (SC31) showing the opposite and more canonical phenotype (i.e. tumor promotion by estrogen). They evaluated tumor growth in vivo and in vitro (using organoids) in response to E2 and ER antagonists treatments, analyzed candidate ER target gene expression upon such treatments on mRNA and protein levels, and most importantly used bulk and single cell RNAseq approaches to understand the global changes in transcriptomic profiles in these two phenotypically different PDX models upon E2 treatment. The most unique aspect of this study is the unbiased characterization of the transcriptomic profiles of ESR1+ vs. ESR1- cells co-existing in the same tumors, which according to the authors have not been shown previously. By doing so the authors made the interesting observation that ESR1- cells in the tumors are regulated by estrogen and showed overlapping transcriptional changes as ESR1+ cells. Overall, this is an interesting study probing for mechanistic insights for a poorly understood yet clinically relevant question. The paper is clearly written and data largely support the conclusions.

Major comments

  1. In Figure 1b, please include quantification besides the representative images.
  2. In Figure 1, the authors indicated in the legend that E2 was administered differently for the two PDX models (together with tumor cell injection or after). Please address this in the results section and explain why.
  3. In the beginning of the results section, the authors described that genome sequencing revealed no ESR1 amplification for GS3. However, it would be important to check ESR1 protein level also to determine whether it is significantly higher than SC31. The anti-ER alpha IHC images in Fig.1b suggested this could be the case but without proper quantification it is difficult to tell for sure. Western blot would be better to compare ER alpha levels in the two tumors. This is an important point to address since high ESR1 protein levels, whether due to gene amplification or other mechanisms, has been shown to cause estrogen-induced tumor suppression. If observed, this could change the conclusion of the paper.
  4. It is difficult to understand why MMP counteracts the tumor-suppressive effect of E2 in GS3 (Fig. 2b) whereas ICI shows the same effect as E2 and did not affect E2 effect when combined (Fig.S8), and the authors did not provide any explanation. Though differing in mechanisms of action, both drugs inhibit ESR1 function and should theoretically produce similar phenotypes. The authors should at least compare the two drugs side by side in vitro for their effects alone on GS3 and in combination with E2 (as in Fig.2b). Conceptually, the authors need to discuss why similar antitumor effects of E2 and ICI are expected in Fig.S8, given their contrasting effects on ESR1 activity.
  5. It is difficult to comprehend the model in Fig.7 as it is currently drawn. Please improve the clarity of the figure as well as the figure legend to better relate to the key findings of the paper and the overall hypothesis.  
  6. Given that the most interesting finding in this paper is the E2 regulation of ESR1- cells and its potential contribution to the tumor-suppressive effect of E2, the authors should provide a proper discussion or speculation as to how this may occur mechanistically.

Minor comments

  1. Please include references for the last sentence of the first paragraph of Introduction (i.e. clinical trial studies showing therapeutic effects of estrogen).
  2. Lines 34-35, grammatical issue of the sentence needs fixing.
  3. Line 54, the word “receptor” is missing (after estrogen).
  4. Lines 107-108, please describe in the results and corresponding figure legends that GS3 did not grow in ovariectomized mice.
  5. Lines 414, the word “by” seems out of place.
  6. Lines 426 onwards, Fig.S7 was mistakenly referred to as Fig.S8 many times. Please check this throughout the paper.
  7. Title of figures 6 and 7 seem to be mixed up.
  8. Figure 6, please indicate in the legend that GS3 was used.
  9. Line 445, extra space was in the word “Although”.
  10. Lines 474, the word “which” should be replaced by “whose”.
  11. Line 499, the word “in” seems out of place.
  12. Line 505, figure S9 should be figure S8. Also, please indicate in Fig.S8 legend what statistical test was used.
  13. Line 520, Figure S10 should be Figure S9.
  14. Lines 522-523, if the authors wanted to discuss the association between IL24+ and NFKB pathway in the context of endocrine therapy resistance, please be more specific and provide a logical speculation.
  15. Line 535, the word “expression” (after gene) should be deleted.
  16. Lines 526-564, the statement seems premature and the authors should acknowledge that the existing data in this paper are purely correlative. More functional studies are needed to determine whether changes in IL24+ (and other genes) are causally linked to E2-induced tumor suppression.

Round 2

Reviewer 1 Report

The authors addressed all my concern.

Author Response

Dear Reviewer 1,

Thank you very much again for carefully reviewing our manuscript (cancers-1493701). The manuscript has been carefully checked English typos.

Best Regards,

Hitomi Mori

Reviewer 2 Report

The authors have addressed the major concerns and improved the clarity of the paper. There are a lot of interesting correlative findings which warrant future mechanistic characterization. The graphic abstract can still be improved to convert the central message. For example, the way it is currently drawn makes it look like NFkb pathway is only active in ESR1- cells whereas the trancriptomic data in the paper clearly showed that this is not true. 
